# Driving Forces and Barriers for the Implementation of Mobility Services in Austria—A Practitioner Perspective

**Marc Schabka [1],\*, Aurelia Kammerhofer [2],†, Valerie Batiajew [3] and Maria Juschten [3]**

[1] Institute for Ecological Economics, Vienna University of Economics and Business, 1020 Vienna, Austria
[2] Institute for Spatial Planning, Technical University of Vienna, 1040 Vienna, Austria
[3] Institute for Transport Studies, University of Natural Resources and Applied Life Sciences, 1190 Vienna, Austria
\* Correspondence: marc.schabka@wu.ac.at
† These authors contributed equally to this work.

**Abstract:** There is growing interest in the role of integrated mobility services in successfully transforming mobility systems by improving alternatives to individual motorised transport and thus contributing to a reduction in negative impacts on nature and society. This paper analyses the conditions for the successful implementation of local mobility services in Austria by adopting a mixed-methods approach combining grounded theory and critical realism. In total, 15 narrative, semi-structured interviews were conducted, sampled by applying thematic, practical, and criterion sampling and using an analytical procedure of open, axial, and selective coding. Particular attention was paid to the influence of governance structures and related processes, the integration of user needs, and the topic of technology and its role in local mobility services. The results showed that for the success of local mobility services, in addition to the availability of and long-term commitment to funding, the initial phase of a project (e.g., the selection of project partners) and the deployment of collaborative and local participatory target-setting processes are extremely important. Further, the findings showed that the motivation and interest of all stakeholders involved in the projects can be regarded as driving forces for enhanced cooperation, dedication, and resilience throughout the project. In addition, developing and communicating needs-oriented mobility services should be emphasised. Although the analysis showed the importance of the deployment and hence the selection of certain types of technological solutions, it also highlighted the difficulties of governance processes related to choosing and implementing adequate technological solutions regarding cooperation and networking between key stakeholders in the projects.

**Keywords:** integrated mobility services; local mobility projects; Austria; critical realism; grounded theory; governance

## 1. Introduction

The transportation sector is contributing significantly to the climate crisis due to its high share of GHG emissions, with transport-related activities currently accounting for almost a quarter of worldwide emissions [1]. Researchers emphasise the need for effective action to limit the environmental impact of transport activities [2] and demand a transition in transportation planning and practices [3]. Besides the environmental concerns, other mobility-related challenges exist, such as the need for better accessibility in a more sustainable way for ageing societies and especially in rural areas [4].

Recent developments in the field of technology and digitalisation have enabled new forms of mobility services, such as ride-hailing and ride-sharing platforms [5,6], that could reduce transport's GHG emissions and improve accessibility. New technologies such as autonomous driving and intelligent transport systems (ITS) are expected to have a high impact on future transport supply, demand, and related planning processes [7].

In Austria, several local mobility strategies have emerged in recent years that have tried to tackle environmental and accessibility-related challenges by providing novel mobility services [8]. Examples include local car-sharing schemes, on-demand bus systems, autonomous systems to cover first- and last-mile problems, and private ridesharing schemes. Most of these are implemented by municipalities, whether alone or in cooperation with several others, predominantly in rural or peripheral areas of Austria, where traditional public transport systems are difficult to operate with cost-efficiency in mind.

The local, small-scale implementation of such smart mobility solutions faces several challenges, especially in rural settings. For one, public funding represents a driving force of transport innovations. However, the dependency on this type of funding is a major issue that can function as a barrier [9]. Previous studies [8] have already highlighted the struggle to secure long-term funding as a pivotal factor in the implementation of mobility service projects. The lack of sustainable implementation and funding schemes causes many projects to disappear after the initial external funding runs out [8,10]. Besides funding issues, the literature mentions a range of other political, institutional, cultural, and economic barriers [8] also concerning user perspectives [11–14]. The transformation of users' mobility habits [12–14] and asymmetries between their (mobility) requirements and the services offered [14] are seen as major barriers that could be tackled through new (information) technologies [13]. Furthermore, participatory processes with all relevant stakeholders, for example, have been identified as a crucial factor for the successful implementation of mobility projects (see [15–17]), highlighting the importance of governance-related processes [18]. Moreover, governance-related processes are seen as interrelated with the technological challenges of mobility services, for example, concerning big data solutions [19].

These different driving forces and barriers for new (technology-driven) mobility solutions are systematised differently in the current literature, for example, as (1) institutional factors concerning users or citizens, governments, and mobility service providers [14,20]; (2) supply-side and demand-side barriers concerning governance processes, legislative frameworks, and the habits and requirements of different target groups [21]; or (3) barriers concerning users and governance and barriers concerning information and communication technology (ICT) [7]. This has led to the understanding that user perspectives, governance processes, and technological factors are crucial aspects of mobility services that are highly interrelated and can either enable or hinder successful implementation.

Hence, the first goal of our research was the identification of the governance-, user-, and technology-related driving forces and barriers in the successful implementation of integrated and sustainable mobility projects on a local level in Austria from a practitioner's point of view. The second goal was the identification of fortifying or weakening mechanisms affecting these drivers related to governance, users, and technology. Derived from these research objectives, we aimed to answer the following research questions:

— What are the drivers and barriers (as well as underlying mechanisms) in the successful implementation of local integrated and sustainable mobility services from a practitioner's point of view?
— What measures are applied to overcome these barriers?

The results were based on qualitative narrative interviews and expert workshops with selected project partners from local mobility projects in Austria. Based on this research, we developed a process evaluation tool, which aimed at supporting stakeholders in the process of implementing local mobility projects in Austria.

The paper is divided into four main parts. The first part (Section 2) provides a brief introduction to the conceptual frameworks of the three pillars: governance structures and related processes, user integration and related needs, and the topic of technology and its role in mobility projects. The second part (Section 3) shows the methodology applied and how a mixed approach of grounded theory and critical realism was used. Finally, the third and fourth parts present the results of the qualitative analysis and discuss their relevance in relation to the existing literature and knowledge for the implementation of future mobility projects.

## 2. Conceptual Foundations and Analytical Framework

Mobility innovations consist of at least three interrelated aspects—their technological character, the user perspective, and the pillar of governance [19,22]. Thus, barriers and drivers are found not only in the technological aspects but, to an even greater extent, also within the sphere of user needs or governance aspects [7,12,23]. In this context, existing governance structures and routines of social and mobility behaviour are relevant factors [5,14,21]. Governance may foster or hinder the development of certain technologies, but it also has to react to and cope with emerging transport and mobility technologies in terms of regulation, adoption into existing systems, and their interaction with the built environment [7,21,24].

### 2.1. Technology

Technologies in the transport sector are developing and emerging very rapidly [25]. Current trends in technology are manifold and diverse. On the demand side, they include digital technologies such as reservation and booking platforms and on-demand allocation for on-demand mobility that can be booked from anywhere. On the supply side, they include new propulsion systems, automated driving and within-distance measurement, sensor technology, and communication technology (see [21]). It should be noted that some innovations are incremental, whereas others are disruptive and can have a drastic impact on our society, living spaces, and mobility behaviour [25], as historical developments such as the invention of railway systems or motor vehicles show [26]. Automated driving, as well as Mobility as a Service (MaaS), are examples of current technology innovations that have the potential to be disruptive [23]. Further trends include Shared Mobility and e-Mobility. Mobility as a Service (MaaS) is a trend that builds upon these innovative trends. By integrating technological aspects and considering different institutional settings, routines of mobility behaviour, and user perspectives, MaaS contributes to promoting the acceptance of mobility offers by users and is decisive in terms of the scalability of mobility services [27]. Under the paradigm of MaaS, different demand-side and supply-side technological trends interact with the intention of achieving multimodal, seamless, on-demand mobility [23,27].

These technological trends consist, on the one hand, of the technical integration of offers, booking, and reservation platforms for different mobility services, including sharing offers. On the other hand, relevant technologies are automated vehicles with alternative forms of propulsion. Thus, the trend of MaaS is highly interrelated with the diffusion of internet-connected and portable devices, such as smartphones [23]. This demonstrates that technological innovations need to be seen in relation to other (technological) innovations and their context of application.

The possibility of linking existing services and integrating them into (digital) platforms shows the increasing importance of technology in mobility projects [5,23]. However, this process of integration underlies various technological and governance-related barriers, such as connection via various technical interfaces; legislation and regulations on a national, regional, and local level; and (the lack of) collaboration experiences and the willingness to link services and integrate them into platforms by mobility service providers [12,14,19].

The choice of the appropriate technology can therefore be crucial for the success of a project, but only under further consideration of governance structures and user perspectives. Thus, there are several criteria in the choice of suitable technology, including the purpose of use, functionality, desired user experience, scalability, regulations, financial aspects to be considered, and information about the different technologies to be obtained [22]. Within this process of choosing suitable technologies, networking and strategic cooperation with other mobility service and platform providers are of particular importance to achieve successful, integrated, and connected mobility services [19].

While new forms of propulsion do not change the existing mobility system much, innovations in the field of automated driving can often have a restructuring effect. How far-reaching the effects are depends on the stages of automation and the legal and organisational framework that will be set up for their application. Technological aspects

of automated driving concern actuators, i.e., the control of the vehicle (steering, braking, accelerating); sensor technology to detect the vehicle environment; artificial intelligence and "machine learning" to understand traffic situations and react accordingly; and the further networking of vehicles with each other [28,29]. Practical implementation opens up questions such as driving safety and liability issues as well as the extent to which automated driving can be applied to existing road infrastructures and in mixed traffic. This shows how technology is embedded in institutional and legal framework conditions and related to the context of application and user needs [19,28].

*2.2. Governance Structures*

As existing definitions of sustainable transportation development only focus on the actual outcomes of the process without considering procedural aspects [30], this section aims to shed light on the often-undervalued role of governance processes within the implementation of mobility projects at a local and regional level.

The term governance, which has different definitions across the literature, can be described as "how do, and can, the public sector (government), the private sector (firms and individuals as consumers) and civil society (e.g., non-governmental organisations, NGOs) work together towards [sustainable development] goals" [31]. Besides the existence of numerous definitions across the literature, the term governance is also seen as addressing different dimensions; as Gudmundsson et al. [32] suggest, governance refers to a system of rules (e.g., laws and regulations), networks of actors, and the development of policies themselves to achieve goals. This understanding forms the basis of our subsequent statements.

Taking a look at the current literature regarding driving forces and barriers in the implementation of local mobility solutions, technological aspects have been regarded as less important for flexible mobility solutions [8]. Instead, e.g., flexible micro-transport services face institutional, cultural, and economic barriers [8]. Furthermore, the findings of Baumann and White [30] related to policy implementations explicitly pointed toward governance-related barriers. The implementation of transport policies in a local urban context often faces barriers due to fundamental differences in stakeholders' interests and values. Additionally, Tormans et al.'s [33] findings related to local policy making pointed in a similar direction, as conflicts between local officials and local politicians caused by different visions and interests combined with feeble internal communication hamper local mobility policy making. In the same way, they noted that a "lack of shared objectives between partners in major projects may also lead to problems" [34]. Overall, the need for the transparency of partners' expectations throughout the development and implementation process of the mobility solution, however it might evolve, was pointed out by Jokinen et al. [8]. Apart from different visions and interests as potential barriers, Tormans et al. [33] stated that the enterprising spirit and motivation of all stakeholders as well as communication between the local administration and residents benefit mobility policy making. After all, Baumann and White [30] revealed a connection between collaborative processes and positive outcomes, which underlines the importance of governance processes. Overall, the mindset of involved stakeholders, transparency about common goals and visions, and proper communication appear to act as a hampering or driving force for the implementation of mobility projects. Apart from this, the question of decision-making power became a very important factor in the case of a ceased mobility solution in Helsinki [8]. Furthermore, the low competence of local officials in smaller municipalities can prevent them from elaborating on the required objectives and providing concrete actions in response to local mobility needs [33].

Taking the above into account, in the face of the emerging transition towards MaaS, the need for new forms of collaboration to connect different regional mobility systems has been pointed out [35]. Apart from the topic of governance processes, the cross-sectional matter of governance and technology is also highly relevant for MaaS: not integrating

ticketing for a niche mobility solution with existing public transport appears to be a barrier and needs special attention while implementing a mobility project [8].

Comprehensiveness; the transparency of expectations, visions, and interests; the motivation, competence, mindset, and communication style of stakeholders; finances; and the integration of local mobility solutions into other mobility solutions can be considered as key factors that affect the implementation of local mobility solutions. Nevertheless, Kindhäuser [36] empirically observed that not only the framework conditions of municipal transportation policy but also both the cultural context and urban characteristics vary across cities. These insights indicate that no generally valid governance strategies for success can be derived, which implies that the following findings act as guidance and, if deployed in a local context, need to be adapted to fit the local requirements. Therefore, Kindhäuser [36] mentions the importance of successfully communicated interests and having individuals with charisma and personal persuasiveness in leadership positions as promoters. Finally, flexible structures oriented towards dialogue and consensus; the project-based orientation of the organisation; communication; and external factors such as regional political guidelines, financial support, and the role of pilot projects are further success factors for the implementation of mobility projects.

### 2.3. Users and Markets

This section specifically addresses the existing research on the role of integrating user needs within the implementation of small-scale, local, and sustainability-oriented mobility projects.

When aiming for the transformation of mobility practices, a singular focus on implementing new technologies or services is not sufficient [37]. While there is certainly a growing need and demand for sustainable and integrated mobility services [21], many small-scale implementations of on-demand buses or car-sharing systems fail to meet the needs of potential customers and therefore do not exploit their potential. Besides more "rational" user needs, Sopjani et al. [37] also highlighted the importance of behavioural aspects, including established mobility practices and cultures. To overcome such misalignments of technologies and user needs, various studies have suggested that intensified user involvement in the design of mobility services helps to account for specific mobility needs and practices, local mobility cultures, and available infrastructure [37]. Ultimately, such collaborative design processes are expected to enhance people's travel experience [38], therefore increasing user acceptance and, as a result, contributing to increased willingness to use new products and services [13,39], which ultimately accelerates market penetration [37]. User acceptance is particularly critical when dealing with new, unknown technologies (such as automated vehicles), where safety, as well as ethical and environmental concerns, represent important factors [7,37].

Within the process of user involvement, Sopjani et al. [37] provided an overview of different perspectives on users and their role in the design process, especially for sustainability transitions. This can range from the more passive role of mere consumers (included, for example, through user satisfaction surveys (see [40]), to more active roles as cocreators or even designers and change agents, who lead innovation processes by alternating their routines [41]. Especially within the development of technological innovation, the classic roles of users as consumers have been replaced by more active roles that acknowledge the agency, creativity, and self-awareness of users. From a user's perspective, active involvement can fulfil different functions [42], including the desire to have control over design processes and share personal experiences and knowledge and the ability to bring about change on a larger scale, which may be particularly relevant for sustainability-oriented services.

However, integrating users in the development of entirely new technologies also poses a challenge due to the limitations of people's imaginations [38,43,44], as well as their existing routines and mobility practices [37]. People may not believe that they will need or use something until it is implemented in their surroundings. This limitation can be approached in different ways: in procedural regards, Sopjani et al. [37] concluded that it

is crucial to involve users in the late stage of implementation when true preferences and behaviours become visible and critical mass is needed. For other types of projects, however, the early involvement of users may be critical to allow for enough time to integrate their design preferences and expectations [44]. Furthermore, Sopjani et al. [37] highlighted the importance of including users with different levels of technology affinity and usage likelihood. In methodological terms, researchers from Singapore have addressed this limitation by applying various virtual-reality and simulation-based methods to make new technologies more tangible [38]. In doing so, technological innovation has enlarged its function from the central feature/product of needs-based or human-centred design to an enabling factor to improve the understanding of human needs. Another critical consideration is the diversity or heterogeneity of user preferences and needs, leading to contrasting outcomes regarding technological and procedural needs [37]. One example related to AV is the value of human drivers [38]: for some, this value may be very functional and therefore replicable by specific machines or buttons; for others, it may be inherently psychological or emotional, and so cannot be easily replaced. Albeit focusing on a different mobility service (a sharing service for electric vehicles), Sopjani et al. [37] faced the same challenge within the implementation stage. They addressed it by investigating various user roles in user involvement processes and identifying their specific motivations to participate, analysing their respective engagement intensity and investigating the impact they may exert on other users. Their results showed that active engagement in the design process was also followed by a strong sense of commitment to changing individual practices and, consequently, exerting active influence on other potential users. Overall, their study suggested deploying a "convergent activation strategy" in which a collaborative lab setup, including both users and non-users, is used to initiate a critical reflection on established mobility practices and therefore fosters sustainable behavioural shifts [37].

Butler et al. [21] performed a detailed literature review on barriers to the implementation of MaaS systems. Their review found several user- or demand-related barriers. As suggested by the results of other studies on the barriers to the adoption of MaaS services [13,14] or autonomous vehicles [7], similar categories of barriers (or potential drivers) may apply to the implementation of various forms of innovative mobility services. These categories include different aspects, such as financial (i.e., willingness to pay, preferred subscription types); safety concerns (especially related to AV and shared mobility services, see [7]); legal concerns (i.e., data protection concerns for MaaS services, see [21,45]); socio-economic and demographic aspects (i.e., the influence of age, gender, and car usage); design aspects (i.e., possibilities for incentivising usage through personalised features); technological aspects (i.e., possibilities for route optimisation and real-time travel information, see also [13]); functional aspects (i.e., the awareness of personal advantages/lack thereof with regard to convenience, comfort, etc.); cultural/social aspects (i.e., the traditional role of and associations with private vehicles); emotional aspects (i.e., trust/distrust, fun, fear, low inclination to try out new technologies, etc.); and market aspects (i.e., mismatch between targeted and reached users).

## 3. Research Methodology

### 3.1. Research Design—Case Study Approach

A qualitative research design was chosen to understand the processes of developing and implementing mobility projects. Thus, the main research aim was to identify the driving forces and barriers related to governance, technology, and users' perspectives, as well as strategies to overcome them. An empirical, inductive approach was chosen using different qualitative methods to collect, analyse, and reflect data.

The research design relied on "narrative interviews", as they provide an in-depth understanding of chains of events and actions by enabling the interviewee to speak freely without interruptions and to choose relevant events and content [46,47].

In total, 15 narrative, semi-structured interviews were conducted in spring 2020. The sample was defined by applying thematic, practical, and criterion sampling (see

Appendix B, Table A1) while screening for Austrian mobility platforms and databases. The interviewees were selected according to their role within the projects of the defined sample, and therefore not by considering their demographic characteristics.

The methodological approach of the data analysis was chosen by laying out the requirements for the analysis based on the research questions and the narrative interview approach:

- Applying an explorative, inductive approach to identify and understand patterns, relations, processes, and consequences.
- Applying a problem-oriented, practical research approach to focus on driving forces and barriers.
- Ensuring that the deep information gathered by narrative interviews is not lost within the process of analysis [46].
- Comparing across datasets to identify commonalities and differences and build categories [46,47].

According to these requirements, a cross-case analytical approach (see [48]) based on the triangulation of methods of grounded theory (GT) and critical realism (CR) was seen as the most suitable approach. Using an analytical procedure of open, axial, and selective coding [49], GT was used to elicit recurring themes, patterns, and contexts across all projects and to identify and understand (key) categories. While GT represents an open, inductive research style aiming to explore and solve problems [50], the use of a critical realist methodology is deployed to obtain an even more in-depth picture of the underlying mechanisms at work. Therefore, the goal was to use GT and CR's compatibilities to surpass a merely descriptive representation of the results but also to approach them with preconceived analytical concepts such as emergence and generative mechanisms [51].

The analytical framework described in Section 2 was used for precise research questions and subsequent questions in the interviews conducted. Furthermore, the literature was used to reflect and validate the results of the analysis [49]. However, the literature on GT points out that prior knowledge and referring to existing theories within the analytical procedure might hinder the exploration of new patterns [52,53].

After synthesising the findings of the critical realist analysis and GT, an additional literature review was conducted (see Figure 1). The results were processed, visualised, and reflected in workshops with former interviewees and external experts.

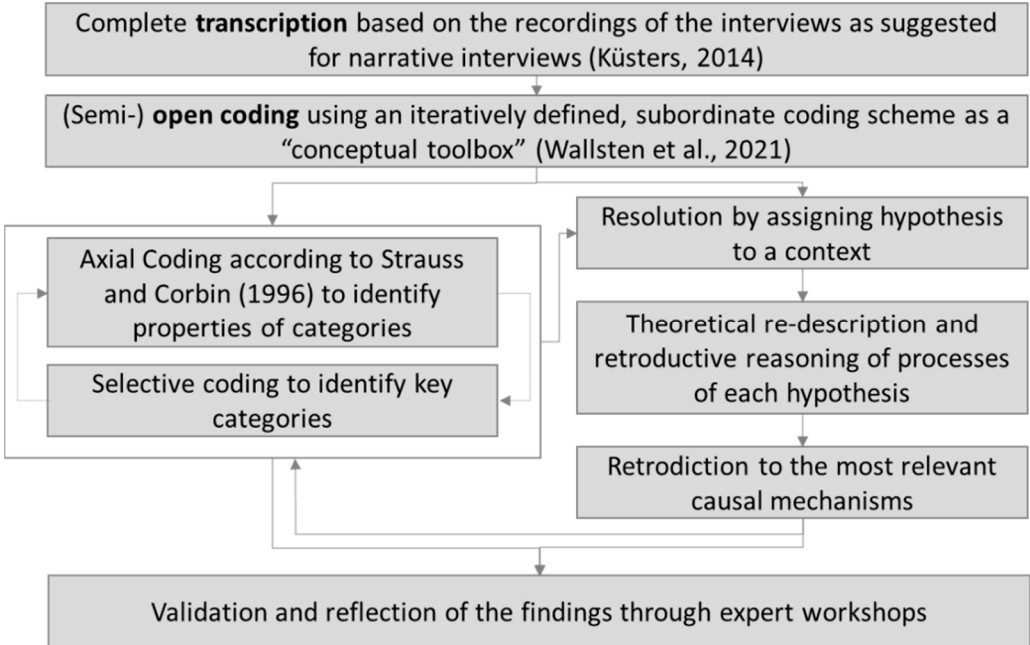

**Figure 1.** Main analytical steps of the process [47,49,54].

*3.2. Data Collection and Interview Conduction*

While the "quest for drivers/barriers or influence factors" could involve a quantitative approach that asks for pre-defined "success factors", a narrative, process-phase-oriented approach was conducted. Based on "narrative theory", open narration enables the reproduction of processes and decisions that lie further in the past [47] and thus captures experiences [47]. The literature shows that narration represents processes of action and chains of events structured by the interviewee as a temporal and thematic storyline [46,47], while other research methods might lead to the listing of events. This enables the discovery of subconscious factors of influence and enables interviewees to speak more openly about problematic project phases or influencing factors [46]. This allowed us to systematically gather information about the course of action, crucial decision points, and the relevant stakeholders involved within these steps, to understand the (temporal) relationship and resulting dynamics between these aspects.

To achieve these aims, certain methodological requirements needed to be considered. Establishing trust at the beginning of the interview was very important, as open narration relies on trust and a comfortable interview situation. The researcher should be able to listen, encourage the narration, and avoid interruptions while taking notes for subsequent questions [46]. Thus, each interview was conducted by two interviewers (*A* + *B*). Due to the COVID-19 pandemic restrictions, most of the interviews were carried out online via videocalls.

The interview was structured as suggested by the literature [46,47], starting with (1) an introduction as a stimulus to enable the interviewee to speak freely and (2) a free narration of the project process and consecutive questions on the decisive factors of the process to deepen and clarify the interviewees' answers. This was followed by (3) a question to cocreate a "project timeline" using the online visualisation tool "*Miro*" (see Appendix A, Figures A1 and A2) and (4) an in-depth focus on individual decisive events according to the free narration. Whereas phases (1) and (2) followed guidelines of narrative interviews [47], phases (3) and (4) were similar to a (semi-) structured interview and concentrated on further specifying segments of narration and bridging gaps [46,47].

*3.3. Data Analysis Based on a Grounded Theory Approach (GT)*

GT is seen as a comparative, pragmatic research style [50] that enables problem-solving research by observing processes and their results [52]. Furthermore, GT shows practical explanatory potential, leading to a deeper understanding of actions and processes [52]. GT represents an open, inductive approach that enables the researcher to break through pre-assumptions, decompose collected data and restructure it, and build theories [49].

To avoid the danger of "idiosyncratic aberrations" [52] within the interpretive analysis, on the one hand, the on-going validation of the data is necessary [49]. This is where CR becomes relevant, as it questions mechanisms more deeply. On the other hand, analysis as a collective process [52] and independent reflection by several researchers [46] are beneficial. Thus, analysis was not only conducted in a division of tasks but as joint analytical work by the research team.

The research process according to GT comprises sampling, data collection, transcription, and several steps of analysis as an iterative–circular process [49]. With regard to the temporal and financial framework conditions of a funded research project, only the analytical approach was based on GT. The three-step procedure suggested by Strauss and Corbin (1996) [49]—open coding, axial coding, and selective coding— [50,51,53] was adapted to the research design as iterative–cyclical steps and combined with CR.

As the analysis and coding were conducted by several researchers from the project team, full open coding bore the risk of incomparability, thus hindering a comprehensive and integrated analysis of complex phenomena. Therefore, a subordinate coding scheme, comparable to a "conceptual toolbox" [54], was iteratively defined and mainly based on the interview material.

For the axial coding, the decomposed data were reassembled by applying the coding paradigm of Strauss and Corbin [49] to identify the properties of categories in the openly coded data concerning dimensions, context, intervening conditions, causes, consequences, and strategies. In turn, recognised statements were checked in the data to obtain variations and a deeper understanding of the categories [49].

Based on the results of the axial coding, selective coding was conducted. This involved identifying key categories in line with the research question. These key categories were then related to other categories to form a coherent theory [49,52]. Selective coding was also closely interwoven with axial coding, the steps of CR, and the visualisation and processing of the contents in the form of concept maps in Miro. This made it possible to bring together the results from different analysis steps, integrate content recorded in memos, and show their depth and complexity [49].

### 3.4. Critical Realism and Grounded Theory: Reflection and Synthesis

CR is a philosophy of science developed by Roy Bhaskar in the 1970s [51,55], aiming to move from the realm of prediction towards the realm of explanation in empirical research by targeting not the regularities or patterns of events but their structures and mechanisms [55]. Furthermore, CR's purpose is to enhance a scientific investigation by drawing on the distinction between ontology and epistemology in the realm of reality and proposing a 'stratified ontology', namely that the "conjunction of two or more features or aspects gives rise to new phenomena, which have properties which are irreducible to those of their constituents" [56].

While GT can be regarded as an inductive scientific investigation to systematically develop a theory [51], it encompasses a data collection and analysis process that is compatible with a critical realist methodological approach. According to Oliver [51], "critical realist Grounded Theory would address both the event itself and the meanings made of it, approach data with the preconceived analytical concepts of emergence and generative mechanisms and pursue emancipatory, rather than merely descriptive, goals".

The methodological approach suggested by Bhaskar, which was followed in this study, can be divided into an eight-step approach, summarised by Bhaskar as the "RRREI(C)" approach [57]:

- Resolution, abductive redescription, retroduction (RRR).
- Inference to the best explanation (I).
- Retrodiction, elimination, identification of antecedents, and correction (REI(C)).

The so-called resolution, which refers to breaking down complex events into components [58], was carried out based on the results derived from the open and axial coding process. Each hypothesis generated from the semi-structured interviews was assigned to a context (or category) to "identify and make explicit the connection between concepts and categories" [58]. This was complemented with a literature review to consult the pre-existing theoretical knowledge as a benchmark for the results.

The second and third steps were the theoretical re-description (abduction) and the retroductive reasoning process of each hypothesis resulting from the coding process, aimed at drawing an inference regarding possible mechanisms [58]. The next steps were the inference of the best explanation and the retrodiction of the most relevant causal mechanisms that could have been responsible for the successful (or unsuccessful) implementation of the local mobility project. This included a visualisation through conceptual maps informed by critical realism and selective coding as suggested by Fletcher [59]. The goal was to graphically show the interdependency between project phases and the related mechanisms.

In order to validate the findings, two independent expert reflection workshops were conducted. Each was structured into two sections. First, the participants were asked to rank categories resulting from GT (e.g., a collaborative target-setting process) by their overall relevance and to add further categories if missing. In the second part, an in-depth reflection on the conceptual maps of the key categories was performed by showing and explaining the visualisations to the participants and by discussing the discovered mechanisms. The

participants were continuously asked if the structures and relations appeared correct, complete, and logical to them and if they were able to identify their projects within the findings. Thus, further aspects and relations could be added and adapted.

## 4. Results and Findings

### 4.1. Particular Critical Thematic Complexes for Successful Implementation of Mobility Projects

In the course of the analysis, key categories were identified that have a potential impact on the implementation success of mobility projects. These key categories were structured along three dimensions (governance, users, and technology). However, the key categories did not stand alone but were interlinked in many ways and interacted with each other. The relevance of these key categories was assessed based on (i) the interviews, i.e., the frequency and the level of detail in which these topics were reported (also with contradictory statements); and (ii) the assessments of knowledge gaps and contradictory statements using the CR approach. For further processing, (iii) internal reflection was also carried out as part of the conceptual mapping process with regard to the possibility of proactive action strategies. Based on this, and together with experts and interview partners, we elaborated, reflected on, and supplemented four thematic complexes related to the selected key categories. Possible strategies for action were then derived:

- Initiating a mobility project and choosing a project's partners (Section 4.2).
- The collaborative target-setting process (Section 4.3).
- Considering motivation and interests as driving forces for good cooperation, dedication, and resilience (Section 4.4).
- Developing and communicating needs-oriented mobility services (Section 4.5).

### 4.2. Initiating a Mobility Project and Choosing Project Partners

The interviews and workshops revealed the need for committed stakeholders who take care of the mobility project from the start and throughout the whole process. Therefore, the identification of suitable project partners is essential. The challenges and strategies for identify such partners are discussed in the following section.

#### 4.2.1. Starting a Mobility Project and Keeping It Going

The interviews disclosed the need for stakeholders who push for a mobility project in the beginning. On a local level, these stakeholders are often intermediary actors (such as mobility managers, model region managers, and club chairmen) who are interested in the project itself and are experienced in the field of mobility. Often, these stakeholders have to moderate, balance different factors, and provide decision-making authority. They often act as role models as well.

Not only in the beginning but also during implementation, there is a need for project stakeholders who push the mobility project forward: the interviews revealed that political decision makers (e.g., mayors) who face a particular problem regarding mobility in their scope of responsibility and the "active" attitude of local authorities towards a mobility project can support its implementation [33].

More generally, the attitude and mindset of all cooperation partners can crucially affect the project in a positive or negative way, which is linked to motivation and interest in evolving current mobility offers (see also [30]). The interview partners also highlighted that the mutual commitment and trust of all stakeholders involved form the basis of the progression of the mobility project, which the workshop participants confirmed. The interview and workshop participants made it clear that the attitude and mindset of stakeholders at the management level or in positions of power can act as a driver or barrier, which implies a certain dependence upon power relations. One workshop participant underlined this by stating that the boss of a cooperation partner has the power to drive a mobility project by being committed and providing good leadership qualities (see also Innes and Booher in [30]).

### 4.2.2. Impacts of Choosing Suitable Project Partners

As seen above, stakeholders who take care of the mobility project are essential for initiating a project and keeping it going. Therefore, knowing the character of the relationships with potential stakeholders and knowing how to team up with them are crucial.

The interviewees indicated that regional stakeholders are important cooperation partners. The results suggested that regional stakeholders bring strong motivation and interest to a project, as they view mobility projects as a contribution to reaching the goal of improving the local quality of life on a professional level, and provide contacts and an institutional and financial background to support such a project.

In general, interviewees and workshop participants share the view that sympathy and a good foundation of trust regarding professional expertise and on a personal level result in good cooperation throughout the project and are therefore factors for choosing stakeholders as project partners. The participants of the workshops also confirmed that mutual trust and sympathy create better cooperation throughout the project, as they result in a more efficient way of working, reliability, responsibility, and commitment. The findings showed that willingness to accommodate each other can lead to higher organisational flexibility, which may improve motivation. Further, more open communication due to trust and sympathy may lead to an exchange of sensitive data. The workshop participants added that knowing each other personally or receiving a recommendation regarding potential project partners from a trusted person is important and that communicating in a "common language" and sharing similar objectives are factors for choosing or rejecting a potential project partner. Baumann and White [30] also found that trust and mutual understanding allow new ideas and technologies to gain ground more quickly and that trust generates more openness and tolerance towards pilot projects.

Not only do trust and sympathy form the basis of cooperation, but sometimes certain project partners need to be chosen because they hold a powerful position within the local context, as one workshop participant revealed. Another workshop participant stated that the certification of potential project partners (e.g., ISO standards) can be a reason for choosing or rejecting a potential cooperation partner. Apart from this, the interviews also revealed that receiving selective assistance from external experts can act as a driver for choosing a project partner for selective cooperation (e.g., legal advice). One workshop participant added that in some cases, competition law can be a hindering factor, as it prevents a certain constellation of project partners.

Regarding choosing political stakeholders as project partners, the interview results and feedback from the workshops diverged. On the one hand, the interviewees assumed that it might be beneficial if political decision makers feel that they have been involved in the decision-making, although, on the other hand, it may also be beneficial for the mobility project if political stakeholders are kept out of the planning process. Adding to this, Baumann and White [30] stated that stakeholders who do not have a formal influence on the decision-making process often contest policy implementation; therefore, the involvement of (political) stakeholders needs to be considered carefully. One workshop participant highlighted that political project partners can hinder the development of a mobility project in cases where they weigh political interests higher than their commitment to the mobility project: *"Political commitment is usually more important than commitment to project partners."* She also adds: *"We have many projects where the mayor is actually behind it. Actually very much welcomes the project, but on the part of the party line, it's just not done."* Raising awareness of this issue and "showing backbone" by implementing the mobility project even without political partners were coping strategies mentioned during the workshop to address these difficulties. In conclusion, the interviews revealed that professional expertise and political forces must interact for the initiation and implementation of a mobility project to be successful.

### 4.2.3. Strategies for Identifying and Choosing Suitable Project Partners

The interviewees revealed different strategies for identifying suitable project partners: clear decision criteria for the selection of project partners are essential, but also the consideration of the long-term willingness to cooperate and the characteristics (moderating, balancing, decision-making competence, exemplarity) of potential driving actors. Choosing partners who are affected by or aware of the challenge that the mobility project tries to address and selecting interface actors and stakeholders with political skills is an option.

Apart from this, choosing partners who are linked to the region or strive for inter-municipal cooperation is considered a driver. Therefore, involving driven and motivated actors who are in personal contact with (political) stakeholders and approaching political stakeholders on a personal level and presenting good-practice examples was another strategy that was mentioned as a successful way to convince actors to team up.

The workshop participants also mentioned that they select partners with suitable competencies and pointed out that federal states can provide helpful information about potential cooperation partners as well as contact networks of stakeholders on a national level (e.g., regarding climate-friendly mobility). They also stated that establishing contact with potential project partners working in institutions is important for forming strategic collaborations and is a good way to identify project partners.

Finally, once cooperation with project partners is set up, recording the commitment in writing and thereby consolidating it (community council resolution or letter of interest) was mentioned as an essential step to confirm the commitment.

In conclusion, our research revealed the following strategies: approaching potential stakeholders on a personal level, contacting networks of stakeholders on a national level, and establishing contact with individuals working in institutions. In more detail, this is achieved by conveying a personal benefit for the potential partners and the presentation of good-practice examples of other mobility projects. Finally, making a binding written record of the commitment and thereby consolidating it is an important strategy once cooperation has begun. To put this in context with the existing literature, Jokinen et al. [8] stated that a signed letter of intent acted as a starting point for launching demand-responsive transport in Helsinki. Adding to this, Baumann and White [30] highlighted the importance of a signed document that determines the principles of consensus *('consensus corridor')*.

### 4.3. The Collaborative Target-Setting Process (CTSP)

In relation to the selection of potential project partners, our research revealed that the target- or goal-setting process at the local level is one of the most relevant factors for the successful implementation of a mobility project. Both the interviews and reflection workshops showed the importance of the collaborative aspects of creating common visions and how to address them, as well as the recognition of its interconnections with other areas from the analysed projects. Target-setting processes, whereby the public and private stakeholders deliberately and collectively work towards a common goal, can be defined, according to Ansell and Gash [60], as collaborative governance processes, the implementation of which, under certain circumstances, leads to successful collaboration.

The interviews and subsequent reflection workshops identified three main successful actions within the category of target-setting processes: a clearly defined project vision and outcome, overall output at an early stage, and clearly specified project sub-tasks to reach the overall output.

### 4.3.1. A Clearly Defined Project Vision and Outcome

An explicit project vision or outcome was associated with increased personal motivation and the identification of all involved stakeholders with the project team. It enhances clarity, transparency, and a shared understanding of the planned intentions of the project, ultimately enhancing cooperation while reducing conflict among the stakeholders.

Further importance was given to the results from a participatory target-setting process for mobility services at the local level (e.g., municipalities), since offering a platform of

discussion to all indirectly and directly affected stakeholders provided faster dissemination of the project intentions.

### 4.3.2. Overall Output at an Early Stage

The establishment of an overall objective or output at an early stage in the process was often claimed to function as a 'project compass'. This step enables guidance during the whole project and is associated with a reduction in frustration and the avoidance of any unnecessary duplication of efforts among all directly involved stakeholders. Furthermore, the identified objective or output would prompt a clear outline of the possible target group to address. An example of an output could be the installation of a battery-powered and shared taxi network in a local community.

### 4.3.3. Clearly Specified Project Sub-Tasks to Reach the Overall Output

The formulation of clearly specified project sub-tasks to reach the overall objective was also associated, similarly to the establishment of an overall objective, with better cooperation among the involved stakeholders in a project, due to the provision of a clear overview of the action steps.

The analysis also showed that collaboration-oriented target-setting processes themselves might have a direct positive or negative impact on other project processes (e.g., internal and external communication and marketing) and might foster better integration between various political and economic interests. The latter was associated with an increase in cooperation and commitment among strategic stakeholders and partners.

Internal and external communication, outlining the technical solutions and features to address the potential output, and defining the addressable target group and the main interfaces of internal and external cooperation directly impact other project areas and decision-making processes.

Fostering the integration of different social, economic, and political interests through a collaboration-oriented target-setting process was associated with a higher willingness to cooperate. "*A shared goal among the partners appeared crucial for the formation of the alliance, which could be called an exploration alliance aimed at learning about the effects of MaaS on the key drivers of the participants [ . . . ]*" [61] (p. 16).

A possible strategy that was suggested included, as a first step, the clear definition of a project vision (or outcome) and an overall objective (or output) at a very early stage of the project, involving all major interest groups and important stakeholders in order to consider different perspectives during the initial phase and trigger better cooperation with strategic partners. A second step should address the clear specification, in close accordance with the most relevant stakeholders in the project, of how to eventually implement the vision or objective by defining sub-tasks and major milestones throughout the project phases.

### 4.4. Considering Motivation and Interests as Driving Forces for Cooperation, Dedication, and Resilience

Within the qualitative interview analysis, dedication, motivation, and interests were identified as key factors for successfully implementing mobility projects. Apart from a clear, jointly defined objective, other important factors are mindsets and personal character traits: across the interviewees, it was considered common sense that motivation and interest have crucial effects throughout the whole process of initiating, developing, and implementing a mobility service. Furthermore, a sense of community and trust play an important role. As such, the conceptual map informed by critical realism and grounded theory analysis highlighted that these factors lead to resilience within the process of the project, good cooperation, dedication, and commitment. In the following section, the selected factors and their influence on various sub-sections of successful mobility projects are discussed.

### 4.4.1. Fostering Dedication and Resilience

Above all, the interview statements highlighted that dedication and resilience are necessary to make concessions, take further steps in the process, and overcome possible hurdles in the course of a project without a negative atmosphere, tensions, or capitulation within the project team. Shared commitment and trust in the project enable resilience and therefore lead to concessions on the part of the participants, such as the acceptance of financial burdens (e.g., workload). The participants of the workshops also confirmed the need for commitment between cooperation partners and trust throughout the mobility project. This trust and commitment are built through cooperation and the knowledge and hope that the project will progress. The interviews also revealed that missing commitments could result in long delays and therefore impede implementation. The inference process revealed that in cases where no commitment is brought to the mobility project, cooperation partners will unconsciously "hold back" and might actively hinder the progress of the mobility project. The interview respondents also designated commitment above political arrangements as essential for cooperation throughout a project and identified a demonstrable demand for the mobility project as a driver for commitment.

### 4.4.2. Addressing Personal and Institutional Interests to Foster Dedication

Institutional and personal interests are interrelated and express the interest of the project participants to participate in the project. As revealed above, a positive basic attitude and interest in the mobility project promotes good cooperation, dedication to the project, and overcoming challenges. Apart from this, the interview results showed that personal interests affect the perception of problems and have a significant influence on a project's stamina and resilience in the face of challenges. Personal interest on the part of decision makers and employees includes concern, self-fulfilment, a positive attitude, curiosity, the desire for the recognition of successes and achievements, and (in particular) idealism, as revealed within the process of retroduction. The workshop participants suggested that opportunities for involvement, codecision making, and self-fulfilment promote personal interest throughout a project. These strategies further affect the sense of community and trust between project partners.

Institutional interest refers to the different political and economic interests of an institution. These political and economic institutional interests can hinder cooperation concerning the balance of power but can also be a main promoting factor. Within the process of retroduction, the distinction between an internal and external perspective was revealed. The internal perspective ties in with personal interest and is characterised by goals, the mission statement, the role of the institution, and concrete tasks. Cooperation, other institutions, and external expectations (e.g., pressure to complete the project by the funding agency) influence the external perspective of institutional interest and political support. Other influencing factors include feedback to the institution in terms of external perception and the institution's attention to mobility-critical issues. Political and economic incentives (e.g., overarching strategies, policy papers, and funding programmes), as well as developments in the mobility market, are external incentives for personal and institutional interest.

To foster these institutional interests and personal motivations, the workshop participants suggested strategic cooperation with relevant actors and partners with high competencies, as well as the targeted monitoring of market dynamics. Further, gathering information on relevant strategies, policy papers, and financial incentives and observing the mobility market and good practice projects are possible strategies for discovering incentives, fostering personal and institutional interest, and gaining relevant knowledge and inspiration for a project.

### 4.4.3. Enhancing Trust and a Sense of Community

Not only personal and institutional interests but also the perception of problems and (in particular) an urgent need for action (feeling pressure) influence commitment to a project and whether the project partners believe that they are working towards a larger

goal and identify with the project. Whereas feeling under pressure was mentioned by interviewees as a crucial aspect, the perspectives of the workshop participants diverged and relativised the extent of this factor's influence: "Some kind of suffering pressure or willingness to change can always be addressed by projects".

Motivation and identification with a project essentially shape the sense of community, as was highlighted by the interviewees and workshop participants. Common personal and institutional interests and jointly defined goals also influence the sense of community. Sympathy, trust on a personal and professional level, and a consensus between the involved project partners are essential for a sense of community, a positive atmosphere amongst the project partners, and further commitment and motivation throughout the entire course of the project to implement the project steps efficiently and cooperatively. A good relationship between the actors further improves the flow of information and enables a high level of the acceptance of joint decisions. Strategies suggested by the workshop participants to promote trust and a sense of community included transparent and on-going communication at eye level and transparent goals, as mentioned earlier.

### 4.5. Developing and Communicating Needs-Oriented Mobility Services

The results of the qualitative interview analysis identified several important categories that can be subsumed into the overall theme of the challenges and strategies related to the development and communication of needs-oriented mobility services. Across most interview respondents, user involvement was commonly believed to be a central aspect of various development processes from service development and refinement to implementation and communication. As such, the visualisation resulting from the axial coding (and critical realist analysis) showed that the use of mobility services depends strongly on people's perceptions of the benefits of using the service as well as their overall acceptance of the service. The subsequent section will discuss some of the factors that influence the extent to which specific mobility services meet these objectives.

### 4.5.1. Enhancing a Sense of Ownership and Identification

Above all, many interviewees shared the view that early user involvement increases the sense of identification between users and a service. When inquiring about the need for a certain level of identification, several interrelated objectives were named, including the feeling of involvement in ownership, which in turn affect communication with other people about the service. This is especially true when personal identification is a result of the mobility service matching one's mobility needs, thereby creating a tangible personal benefit. One participant of the reflection workshop emphasised this by stating, "*If I identify with a shared taxi service because I have often needed it myself, then the way I'll communicate it to others is totally different.*" This is believed to be particularly relevant for stakeholders in political functions, whose outreach is often higher and who can act as powerful enablers or barriers to the success of such projects or services.

### 4.5.2. Developing Mobility Services That Generate a Personal Benefit/Added Value

According to the interview results, the feelings of identification or ownership mentioned above require initial adopters (and subsequent multipliers) to perceive a personal benefit from the use of a service, often relying on real-life personal experience instead of mere information about the service. Therefore, the second objective is to understand user needs in order to generate actual added value through a particular service. In this regard, the interview results were not fully conclusive and diverged depending on the specific context of the interviewees. While some highlighted the role of early (and continuous) user involvement through user surveys, codevelopment workshops, and market analyses, others were more sceptical about this approach. According to one workshop participant, asking people about their needs may even generate counter-productive results due to people's lack of imagination regarding more radical systemic transformations. Metaphorically speaking, he stated that people may wish for "*faster horses*" rather than cars, or, transferred

to current society, they may wish for "*greener cars*" rather than car alternatives. In saying this, the interviewee still highlighted the importance of keeping people's personal benefits in mind. However, instead of asking people what they want, the interviewee suggested focusing on creating a positive, fun, and comfortable user experience that can present an alternative to the comfort that lies in habitual choices. Despite the scepticism about the benefit of user involvement in the early development stages, the workshop participants agreed on the importance of including relevant stakeholders and potential users once the overall design of the service is in place. According to them, "*it is necessary to demarcate in which area citizens can have their say*", a fine line that ideally involves the feedback of relevant stakeholders (i.e., policy makers and experienced experts).

As relevant strategies to foster the creation of personal experiences by incentivising initial usage, the interviewees and workshop participants mentioned positive communication regarding the new mobility services to achieve higher visibility and trigger reflection on the potential personal benefits. This was seen as the basis for potential users' willingness to gain their own user experiences. Further, different forms of use incentives, especially test weeks, free-trial periods, and testing opportunities at local festivals, were suggested by some participants. However, not all agreed on this strategy. To share (positive) experiences and trust in the mobility service, it was seen as necessary to actively include these experiences in (social) media communication and encourage the early adopters to spread the word.

## 5. Discussion

Our findings across the three dimensions of governance, users, and technology discussed in the previous chapter highlighted the most relevant processes that might affect the successful implementation of local mobility projects within integrated mobility services. The implications of these results also have direct effects on policy and implementation measures. Therefore, the goal was to compare these findings with recommended actions from previous studies and derive recommendations accordingly.

### 5.1. Initiating a Mobility Project and Choosing Project Partners

The results showed that every mobility project is embedded in local and regional structures. Thus, the process of initiating them is dependent on available knowledge and resources, as well as institutional relations and dynamics (see [62]). This suggests that there might be a lower potential for (disruptive) mobility innovations in rural or peripheral areas due to the existence of more conservative social structures. To address and shape these structures, innovation ecosystems have been mentioned as a possible strategy to connect the existing knowledge and resources of different stakeholders and foster innovations [63]. This may include living labs such as urban or mobility labs and the development of intermediaries whose role it is to connect different local actors [64–67]. The roles that different actors assume will shape the nature of mobility in the future [54].

In local and regional contexts, the role of municipalities has been highlighted. Khan [68] stated in the context of urban low-carbon transitions that: "*The municipality is only one of many actors in implementation, or innovation, networks and is normally not the main driver. However, cities and local governments can do several things to facilitate and support the development of niche innovations*". However, our research suggested that local authorities play a crucial role in fostering the implementation of mobility solutions in Austria. The scope of responsibility and the "active" attitude of these local authorities towards a mobility project can support implementation [33]. Wallsten et al. [54] argued that local public authorities need to take a stronger leadership approach.

According to innovation ecosystems, collaborative stakeholder dialogues and collaborative target-setting processes are valued strategies for paving the way towards successfully implemented mobility projects on a local level. Further, market actors are ascribed as relevant actors [54].

Initiating a mobility project, as a cooperative innovation process, consists of consensus building and joint learning processes [30,63]. This is supported by both the existing literature and the results of this study. They showed that trust and mutual understanding are key factors in initiating a successful mobility project, as they foster the open development of new ideas, knowledge sharing, and openness towards pilot projects [30]. Our results also showed that strong collaborative alliances between all involved stakeholders are a precondition for their financial commitment.

*5.2. Motivation and Interests as Driving Forces for Good Cooperation, Dedication, and Resilience*

Our results showed that good cooperation between all project partners, dedication to the project, and resilience throughout the project are highly dependent on personal and institutional interests and a sense of community that is built on trust and personal relations between the project partners.

In most of the examined projects, personal and institutional interests influenced the resilience of project participants in the face of challenges. In local contexts, mobility projects often face barriers due to fundamental differences in stakeholders' interests and values. If actors do not see what their concrete contribution to the project can be, this reduces their motivation [60]. Additionally, a "lack of shared objectives between partners in major projects may also lead to problems" [34]. However, the role of personal and institutional interests in successful mobility projects has hardly been considered in the literature so far.

In connection with interest, the suffering of project partners can also be considered a relevant factor for resilience throughout a project. While project results paint a divergent picture of whether suffering is relevant for resilience and dedication, the literature on environmental behaviour shows that [60] personal benefits [69] and identification with the situation [70] are crucial. Furthermore, a sense of community, based on trust, personal relations, and active involvement, appears to be an important aspect of the resilience of all project partners—within the project team, as well as among users—throughout a project (see also [70]).

Both our results and the literature suggested that trust and relational aspects between project partners are key aspects for successful cooperation and a willingness to make concessions and joint decisions [61]. Personal relations [71], communication, and active involvement [15] foster trust generation in a project team. Further, trust is seen as a "control mechanism that facilitates timely information exchange" within the project team [72]. Regarding network governance, Sager and Ravlum [73] also mentioned that trust is the basis for action. Therefore, trust seems to be more appropriate than rational profit maximisation (market governance) or compliance with rules and laws (hierarchical governance) (see [74,75]).

Our results showed that beyond the project context, the public sector can stimulate the interest of local institutions by providing information on good practice and the market, as well as relevant strategy and policy papers. In personnel decisions, for example, regarding the role of the project manager and strategic cooperation, specific attention should be paid to institutional and personal interests. Thus, it seems to be important to identify shared interests and objectives between the project partners and strengthen personal interests through participation, active involvement, and a joint role in decision-making processes in particular [15,71].

*5.3. Developing and Communicating Needs-Oriented Mobility Services*

The results showed that mobility projects often aim to not only introduce new mobility services but also transform mobility practices by introducing these services. Therefore, it is necessary to consider not only technical aspects but also, more importantly, behavioural aspects including established mobility practices and cultures [37]. Further aspects play a crucial role. Instead of aiming for "one-size-fits-all" solutions, the findings showed that focusing on specific target groups by considering their needs and requirements is important for developing successful mobility solutions. In particular, when there is no (personal)

pressure for a potential user to change their mobility behaviour by using new mobility services, both our results and previous studies emphasised the need to create personal benefits for users [69].

Considering 'user experience' as key to successful mobility services, the following aspects should be considered in the process: functional aspects, such as usability in relation to the mobility needs and mobility situations of the target group; the necessary knowledge and requirements for using the service; further emotional aspects; and the joy of use [76–78]. To consider these aspects when developing the technological, functional, and design features of a mobility service (see also [13]), knowledge is required about the target group; their mobility behaviour and needs; and their socioeconomic, psychological, and demographic characteristics [79,80].

Our results demonstrated that this knowledge can be effectively collected through collaborative design processes and user involvement throughout the development and implementation process, such as via cocreation processes, design competitions, and financial ownership. Such collaborative processes may lead to more sustainable solutions in the local context [81]. Although collaborative design processes are expected to enhance user experiences, they are also challenging in terms of resources, the knowledge of how to design the process, the lack of public regulations of open data sharing and funding to encourage open innovation and MaaS processes, and the limited imaginations of people regarding disruptive mobility innovations [38,82]. To tackle these challenges, the results suggested an exchange of experiences with or the active involvement of mobility providers, universities, and intermediary actors such as mobility labs and regional development agencies regarding the procedural and methodological factors of cocreative processes [37,38].

Social identity and a sense of community were also found to be important aspects for enhancing user experience—from the development of the mobility service to a strong commitment to using it. The literature also highlighted the importance of social identity for transforming habits into sustainable behaviour [70]. Positive word-of-mouth communication supports the distribution and marketing of new mobility services.

*5.4. Technology*

In comparison to governance-related aspects and the consideration of user needs in local mobility services, our results showed that technology-related aspects are less important for the overall success of local mobility services. Some years ago, technology was assigned a crucial role in sustainable transport, as new technologies such as more sustainable propulsion systems facilitate more sustainable mobility behaviour without changing routines, practices, or even current transportation systems [15]. However, it has been argued that more radical innovations such as automated driving influence existing systems [28,29]. The development of mobility-related technologies has progressed far, and further technologies such as automated driving, sensors, and connected vehicles are predicted to enter the market in the foreseeable future [21,25]. Thus, our findings and the literature raise the question of how these new technologies should be implemented to support sustainable mobility [15,28,29].

Mobility-related technology developments must always be seen as embedded in their social and institutional context, connected to strategy papers, public (collaborative) target-setting processes, regulations and rules, user acceptance, and further local requirements and characteristics [15]. Nowadays, these governance- and user-acceptance-related aspects are seen as the main barriers to implementing new technologies (especially radical innovations such as automated driving and MaaS), while the technology itself is seen to have diminishing importance as a hindering aspect [23].

These barriers are latent in the case of MaaS: the possibility of linking existing services and integrating them into (digital) platforms shows the increasing importance of technology in mobility services [5,23]. Technical solutions are available to share mobility-related data, connect platforms, and implement functions such as paying and booking. However, our results showed that barriers lie within the lack of willingness to cooperate and share data

and the compatibility of different technology solutions. Harmonising these standards through institutional and governance structures appears to be a relevant strategy to counter these barriers [12,14,19,28].

In conclusion, choosing the right technology is a very important aspect of mobility services influenced by institutional and legal conditions. Due to barriers related to governance and user acceptance, radical innovations are less likely to be implemented in local (rural) contexts. The decision-making process, therefore, requires an exchange with other stakeholders and experts and the provision of good-practice examples.

### 5.5. Collaborative Target-Setting Process

The collaborative aspects of transport planning are forms of participatory governance that are often highlighted in relation to the public–private involvement of such processes (see [16]). As was often mentioned in the interviews, collaboration and interaction between all necessary key stakeholders, including not only policymakers, local authority staff, and local and national governing bodies but also regional transport partnerships, mobility operators (e.g., buses), and transport practitioners within the transport area, are regarded as important [83]. In the literature, the importance of CTSPs is associated with several positive outcomes that go beyond the importance of good collaboration between these stakeholders itself. Sager [73] (p. 285) mentioned, for example, in his systematic review of 62 evaluations of transport policy measures in Switzerland, that a particular focus "must be paid to compromise-finding processes between the participating actors during the planning and implementation phases". The results showed that the integration through CTSPs of often diverging interests can lead to a higher willingness to cooperate; enhance motivation; and therefore foster dialogue, involvement, common goals, and consensual solutions—all factors of success mentioned as highly important in the literature (see [36,84]).

Nevertheless, besides creating advantages that might contribute to successful implementation, it may also create barriers and risks that were not considered in our analysis. These shortcomings could arise in the form of, e.g., higher costs, slower implementation, the potential generation of conflicts between stakeholders, and an increased risk of fatigue among actors (see [85]). According to McAndrews and Marcus [16], providing transportation professionals specific training (see also [85]) to enhance such processes could reduce these risks and guarantee, e.g., access for socially excluded groups who are sometimes not included. What the results also showed is that CTSPs strongly complement the decision criteria for the selection of project partners, as long-term cooperation for both execution and maintenance is often required, and the selection of suitable project partners has been positively associated with a higher level of trust and commitment among all stakeholders.

CTSPs have been regarded as being of equal importance as the availability of resources, and hence funding, for local transport policies (see [83]). This also shows their overall importance in solving important institutional, social, and cultural barriers, as mentioned by Banister [86], and therefore increasing the needed collaboration between local and regional stakeholders [71].

### 5.6. Funding

Both the literature review on major barriers in the implementation of local mobility services and the interview analysis and reflection workshops placed great emphasis on the overall topic of funding, which was not explicitly set out in the findings. The availability of resources is among the greatest challenges faced by local authorities [83]. Without the initial financial means and sustained commitment for funding, local transport initiatives might face, in most cases, closure. The assumption is, among other things, that users are less incentivised to change their mode of transport when alternative commitments are short-term. McTigue et al. (2016) [83] stressed the importance of the availability of resources, as they are usually limited. The interviews indicated that the barriers go beyond the problem of limited financial possibilities for implementation to the pressure of limited financial planning security over the medium and long term. The interviewees

also mentioned the possible barriers presented by a lack of expertise in how to apply for funding and the importance of local alliances to obtain and secure funding. These results are in-line with Binnsted and Branningan [87] (p. 10), who stated: "the resource-intensive nature of bidding for and managing separate funding streams and the lack of available staff and skills are often cited as being significant obstacles to obtaining finance". The literature on this topic is very vast and detailed (see e.g., [86,88,89]) and was intentionally not discussed in further detail in our findings, as the focus of this paper was mainly on the implementation of local mobility services. Nevertheless, the recognition of the regional and national framework conditions around the funding of sustainable mobility solutions is important, as the differences between locally financed, stand-alone projects and overarching (regional or even national) sustainable transport solutions are essential, with many local transport solutions requiring long-term funding.

## 6. Conclusions, Research Limitations, and Outlook

### 6.1. Research Limitations and Strengths

The methodological approach of combining grounded theory (GT) and critical realism (CR) is not entirely new (see [90]), but it has never been applied within transport research. The expectation that this approach would offer a deeper understanding of the relationships and processes of local mobility services was our main rationale for its implementation. The findings provided insights into not only the most relevant project phases and key processes but also their interconnectedness, especially among the governance-related processes. Visually displaying these interconnected mechanisms through the Miro tool made it even easier to analyse and reflect on these findings with the experts in the reflection workshops.

Nevertheless, the critical realist grounded theory (CRGT) approach also posed some constraints. The most significant was that this approach was revealed to be extremely time-consuming, since the elaboration of the findings through the RRRREI(C) method, in particular, involved various inference steps that proved to be complex.

Secondly, also due to the constraints posed by the temporal and financial framework conditions of a funded research project and, moreover, the COVID-19 pandemic, the sampling process did not follow a classical GT approach, which had a negative impact on the screening and selection of projects. As a consequence, it was only after the interviews, and hence the coding and analysis, that we realised that all the research projects (three in total) had to be excluded, because they showed different process sequences and, most significantly, only short-term outcome aspirations in comparison to the local mobility-oriented initiatives. Further, the research focused on local mobility services in Austria. According to Kindhäuser [36], such projects must be investigated within their political, legal, and cultural context to derive findings that are usually not transferable to other political, legal, and cultural contexts [81].

All narrative interviews were carried out online (including video), but this did not pose noteworthy negative effects. The flexibility of the online interviews was therefore considered a positive.

Overall, the CRGT approach offered detailed insights into mobility processes and an in-depth understanding of their promoting and restraining mechanisms. This understanding enabled us to derive measures to better foster successful mobility project implementation.

### 6.2. Conclusions

The present paper aimed to analyse and understand the interconnectedness of governance-related aspects, sustainable technology, and user-oriented mobility services for successful mobility project implementation. This approach revealed the relevant mechanisms of key aspects for successful implementation and provided a holistic understanding of mobility services.

The holistic perspective demonstrated that governance-related aspects are highly interconnected with the processes of developing and implementing (integrated) mobility services in Austria. Aspects such as funding, the legal framework, objectives, and coopera-

tion between partners showed a significant influence on the success of mobility services. A new, key finding was derived regarding project-internal governance in connection with institutional framework conditions (the impact of corporate goals and philosophies and competitive thinking between institutions). This highlighted the dependence of local activities for sustainable mobility and the implementation success of local mobility initiatives on superordinate governance levels.

Further, user involvement was revealed to be an important aspect of successful mobility services, especially in spatial areas and fields of transport, where low economic viability is expected. The processes of user involvement are related to governance structures. Both are, for example, determined by personal and institutional interests, which have not been investigated in the literature as a crucial aspect of mobility services.

Factors such as user involvement, funding, and legal framework conditions also highlighted the need for expert knowledge in local mobility initiatives. Thus, intermediary roles such as (urban) mobility labs and regional development agencies are seen as important stakeholders to facilitate stakeholder networks and the dissemination of knowledge.

Sustainable technology is still seen as an important factor for mobility services. The concern is less about the technical aspects of sustainable transport technologies and more about sustainable implementation, integration in existing systems, and the development of new sustainable transport systems from a governance and user-centric perspective.

To conclude, the COVID-19 pandemic might have momentarily affected the perception of users towards the adoption of certain kinds of public transportation (see [91]). However, we believe that this only partially impacted the final selection of local mobility services, as increased user resilience and the long-term focus of these local initiatives are to be expected.

### 6.3. Further Research and Outlook

The elaborated mechanisms and key aspects of successful local (integrated) mobility services enabled us to derive suitable strategies from a multi-level perspective—from local mobility project initiators to national governments. Based on these results, a self-assessment tool for mobility projects was developed (www.innovationsbarrieren.at, accessed on 4 September 2022). The present paper provided a holistic, multi-level perspective on the successful implementation of projects and revealed mechanisms that should be investigated in more depth.

**Author Contributions:** Conceptualisation, M.S., A.K., V.B., and M.J.; data curation, M.S., A.K., V.B., and M.J.; formal analysis, M.S., A.K., V.B., and M.J.; investigation, M.S., A.K., V.B., and M.J.; methodology, M.S., A.K., V.B., and M.J.; validation, A.K., V.B., and M.J.; visualisation, M.S., A.K., V.B., and M.J.; writing—original draft, M.S., A.K., V.B., and M.J.; writing—review and editing, M.S. and A.K. All authors have read and agreed to the published version of the manuscript.

**Funding:** This research was carried out within the project ULTIMOB and within the funding programme "Mobilität der Zukunft-Nr. 12" (English: Mobility of the Future) provided by the Austrian Research Promotion Agency and the Austrian Federal Ministry for Climate Action.

**Institutional Review Board Statement:** This study was conducted in accordance with the Declaration of Helsinki.

**Informed Consent Statement:** Informed consent was obtained from all subjects involved in the study.

**Data Availability Statement:** Not applicable.

**Acknowledgments:** We would like to thank Alexandra Stickler, Aggelos Soteropoulos, Martin Berger, and Tobias Lindemann for their support and Astrid Gühnemann for comments and guidance that improved the manuscript substantially.

**Conflicts of Interest:** The authors declare no conflict of interest.

## Appendix A

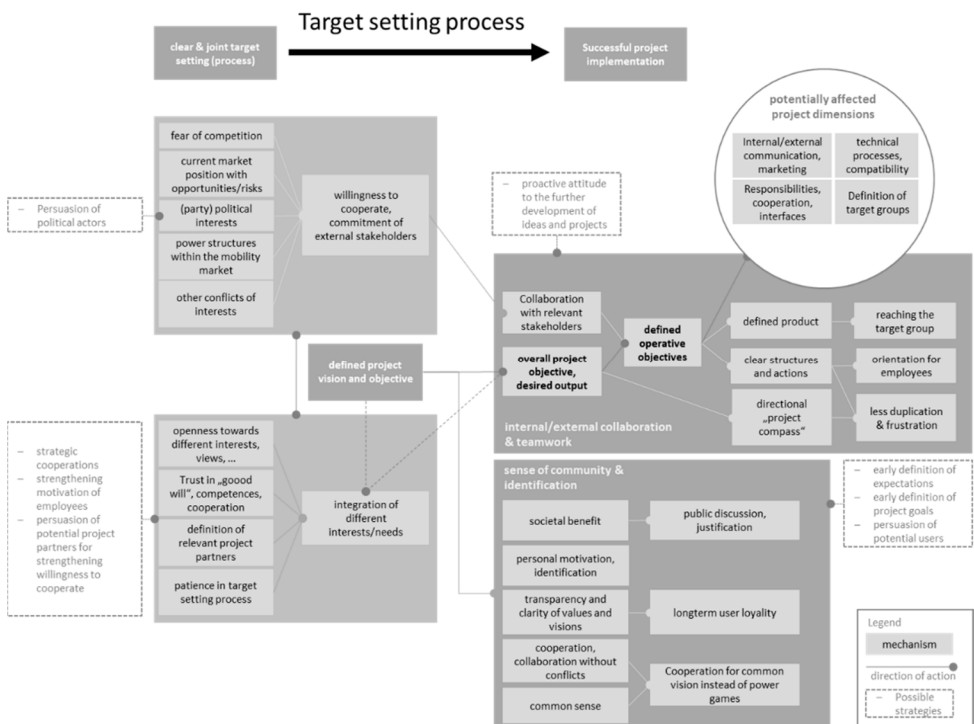

**Figure A1.** Collaborative target-setting process.

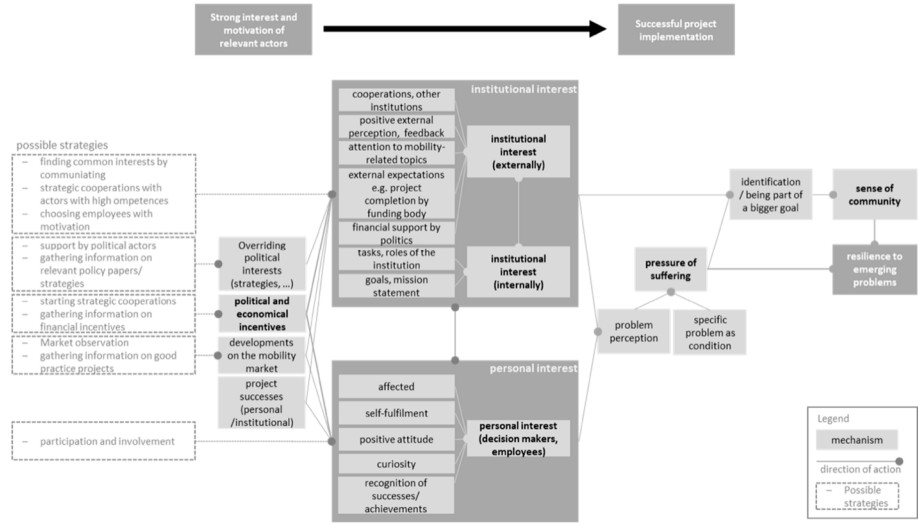

**Figure A2.** Motivation and interests.

## Appendix B

**Table A1.** Sampling criteria.

| | |
|---|---|
| Thematic criteria | − projects relevant to passenger mobility<br>− projects aiming to serve sustainable mobility<br>− projects at the level of prototyping and implementation |
| Practical criteria | − easy access to projects to enable sufficient and in-depth information<br>− recent projects in order to ensure qualitatively rich interviews<br>− in consideration of feasibility and coverage 12–15 interviews were planned |
| Sample criteria | − balance between (research) projects on a prototype level and projects on an implementation level<br>− balance between projects with failed and successful implementationcoverage of as many modes of mobility as possible<br>− coverage as many different occasions of mobility as possible (e.g., commuter, leisure, tourism, etc.)<br>− ccoverage of as many different types of stakeholders as possible (public, private, etc.)<br>− ccoverage of as many different project sizes and complexities as possible (cost framework, implementation period, etc.)<br>− ccoverage of heterogeneous geographical Austrian projects |

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
