# Peer review of "Driving Forces and Barriers for the Implementation of Mobility Services in Austria—A Practitioner Perspective"

_sustainability, doi:10.3390/su141811431_

Round 1
Reviewer 1 Report
Dear Authors,
First of all, many thanks for submitting your paper to this journal and for the time you spent writing that.
The purpose of present paper is to examine the driving forces and barriers to the implementation of mobility services in Austria by using a mixed method approach. There are some comments that will help you improve your research;
1. The number of keywords is more than usual and can be reduced. For example, the mixed methods approach, which includes two methods of grounded theory and critical realism, can be removed or vice versa.
2. In the abstract section; the number of research sample, the type of sampling method, data collection tool, the software used to extract the codes should be mentioned in this section.
3. It is better to present categories and open codes along with their references in an organized table in section 2. Which criteria were extracted from interviews and which through library studies and which from both?
4. In methodology section- section 3; it is necessary to add the demographic characteristics of the interviewees. Also, is a software such as MAXQDA used to extract the codes or not? If used, mention it.
5. The concern about the adopted methodology is related to the qualitative approach. In particular, the review approach described in paper is based on a subjective classification of the papers, that cannot be supported by an objective quantitative analysis but consider several papers "irrelevant" just according to the reviewers' opinion. Therefore, the concern is that different reviewers would classify and select different papers that could lead to different conclusions. The robustness of the adopted methodology should be better explained.
6. In lines 364, 725, 738, ...; it is better to put the reference number after the year. For example, Butler et al. (2021) [18]. Also, in Line 789; The year (2020) should be deleted.
7. Please check Line 434.
8. It is better to use some updated references (2022) in the paper.
9. Please check the text carefully for grammatical errors to improve the readability.
Author Response
Dear Reviewer, (nr.1)
Thank you for the valuable feedback!
We have incorporated you comments as follows:
(1) The nr. of keywords have been reduced (see attached manuscript). NB we have not found where to upload the revised version. That's why the have attached it here.
(2) We have included the requested details (with exception of 'the software used to extract the codes' as no such software was used; but have described it through "open, axial and selective coding" in the abstract.
(3) A table of codes could be added in the annex. As section 2 represents a conceptual foundation based on literature, we would find the codes more relevant for section 3. However, the coding process and the development of the coding scheme are already described in section 3.3: “As analysis and coding were conducted by several researchers of the project team, full open coding bore the risk of incomparability, thus hindering a comprehensive and integrated analysis of complex phenomena. Therefore, a subordinate coding scheme, comparable to a “conceptual toolbox” [36] was iteratively defined mainly based on the interview material.”
(4) As the interviewees were selected by their role in the selected projects, we do not find their demographic characteristics necessary to be displayed in the manuscript (clarified in manuscript section 3). There was no extractions software used (see comment #1)
(5) The methodological approach is based on the research paradigm of Grounded Theory, as described in section 3. The data analysis does not consist of a subjective classification of papers, rather an explorative analysis of interview transcripts. The robustness of the methodology is clarified in the manuscript section 3. Further, in section 6.1 there is additional information over the Limitations & Strengths of the applied methodological approach (GT+CR).
(6) All cited references had been checked again and corrected accordingly (see manuscript)
(7) Done (see manuscript)
(8) We have integrated and updated additional literature undermining our statements (see manuscript and next page).
(9) We have submitted the paper to a proofreading- and editing service beforehand. Nevertheless we have asked a further revision. If some parts/sentences are still unclear please let us know. Thank you.
Further Literature added:
Lucchesi, S. T., Tavares, V. B., Rocha, M. K., & Larranaga, A. M. (2022). Public Transport COVID-19-Safe: New Barriers and Policies to Implement Effective Countermeasures under User’s Safety Perspective. Sustainability, 14(5), 2945.
Flipo, A.; Sallustio, M.; Ortar, N.; Senil, N. Sustainable Mobility and the Institutional Lock-In: The Example of Rural France. Sustainability 2021, 13, 2189. doi: 10.3390/su13042189
Ruhrort, L. Reassessing the Role of Shared Mobility Services in a Transport Transition: Can They Contribute the Rise of an Alternative Socio-Technical Regime of Mobility? Sustainability 2020, 12, 8253. doi: 10.3390/su12198253
Kim, S.; Lee, H.; Son, S.-W. Emerging Diffusion Barriers of Shared Mobility Services in Korea. Sustainability 2021, 13, 7707. doi: 10.3390/su13147707
Storme, T.; Casier, C.; Azadi, H.; Witlox, F. Impact Assessments of New Mobility Services: A Critical Review. Sustainability 2021, 13, 3074. doi: 10.3390/su13063074

Reviewer 2 Report
The topic of the work is very current and will be interesting to the scientific and professional public, but also to business people in the economy.
The introduction is nicely written, but it is too long. Please, try to leave some parts out, especially the parts where you repeatedly cite the same paper/report.
Shorten the title of Chapter 3
The paper is full of technical errors (e.g., the name of figure 1 above the figure itself). Certain sentences should be reformulated.
The methodology of the paper is written in detail.
The most interesting part of the paper is the discussion, while the results are not interesting. Try rewording it a bit.
In conclusion, I would like you to write some practical recommendations.
Although the paper is too extensive, it has the potential to be published. The topic interested me, so I would like the authors to comment in their paper, in a couple of sentences, on how the pandemic (e.g., COVID-19) would have an impact on your topic. For example, papers on similar topics, which I suggest you cite.
Lucchesi, S. T., Tavares, V. B., Rocha, M. K., & Larranaga, A. M. (2022). Public Transport COVID-19-Safe: New Barriers and Policies to Implement Effective Countermeasures under User’s Safety Perspective. Sustainability, 14(5), 2945.
Simović, S., Ivanišević, T., Bradić, B., Čičević, S., & Trifunović, A. (2021). What Causes Changes in Passenger Behavior in South-East Europe during the COVID-19 Pandemic?. Sustainability, 13(15), 8398.
Petrov, A. I., & Petrova, D. A. (2020). Sustainability of transport system of large Russian city in the period of COVID-19: Methods and results of assessment. Sustainability, 12(18), 7644.
The quality of the figures in Appendix A is questionable. I would leave that to the technical editor, but I digress.
Check the way of citing references, it is not consistent everywhere with the instruction for authors.
Author Response
Dear Reviewer, (nr.2)
Thank you for the valuable feedback!
We have incorporated you comments as follows:
(1) The introduction has been shortened. See manuscript. NB we have not found where to upload the revised version. That's why the have attached it here.
(2) The title of chapter 3 has been shortened.
(3) We have checked for technical errors and corrected them. Please let us now where you might see additional ones.
(4) Thank you for the feedback that the methodology is well written.
(5) Thank you for the positive feedback. The purpose was to find a tandem between results and discussion. We would therefore ask if we could maintain the proposed structure. Otherwise this would require a major engagement with the overall chapters (results&discussion).
(6) (a) We have reduced the length of the text by ~1600 words (or 11.5%). (b) We have added a couple of sentences regarding Covid in the conclusion section.
(7) Thanks for the comment. The quality of the Appendix has been slightly improved and could be possibly increased if the technical editor requests it.
(8) References have been checked and corrected.
Further Literature added:
Lucchesi, S. T., Tavares, V. B., Rocha, M. K., & Larranaga, A. M. (2022). Public Transport COVID-19-Safe: New Barriers and Policies to Implement Effective Countermeasures under User’s Safety Perspective. Sustainability, 14(5), 2945.
Flipo, A.; Sallustio, M.; Ortar, N.; Senil, N. Sustainable Mobility and the Institutional Lock-In: The Example of Rural France. Sustainability 2021, 13, 2189. doi: 10.3390/su13042189
Ruhrort, L. Reassessing the Role of Shared Mobility Services in a Transport Transition: Can They Contribute the Rise of an Alternative Socio-Technical Regime of Mobility? Sustainability 2020, 12, 8253. doi: 10.3390/su12198253
Kim, S.; Lee, H.; Son, S.-W. Emerging Diffusion Barriers of Shared Mobility Services in Korea. Sustainability 2021, 13, 7707. doi: 10.3390/su13147707
Storme, T.; Casier, C.; Azadi, H.; Witlox, F. Impact Assessments of New Mobility Services: A Critical Review. Sustainability 2021, 13, 3074. doi: 10.3390/su13063074
